# Do I Really Want to Change? The Effectiveness of Goal Ambivalence Feedback on Dieters’ Motivation

**DOI:** 10.3390/bs12110441

**Published:** 2022-11-10

**Authors:** Javad S. Fadardi, Samiyeh Borhani, W. Miles Cox, Alan W. Stacy

**Affiliations:** 1School of Community and Global Health, Claremont Graduate University, Claremont, CA 91711, USA; 2School of Human and Behavioral Sciences, Bangor University, Bangor LL57 2AS, UK; 3Faculty of Education and Psychology, Ferdowsi University of Mashhad, Mashhad 9177948991, Iran

**Keywords:** health behavior change, change motivation, treatment adherence, goal ambivalence, diet

## Abstract

Becoming committed to a new health-related goal and pursuing it is difficult for many people. The present study (a) developed and tested the psychometric properties of a brief Goal Ambivalence Scale (GAS) in a sample of dieters and (b) tested the effectiveness of providing dieters with feedback on their scores on the GAS. In Study 1, dieters (*n* = 334, 74% females) completed the GAS and a measure of Health-Related Concerns and Actions (HRCA). The standardization of the GAS was supported by CVR and CVI, the results of a PCA, and strong reliability and validity statistics. In Study 2, the experimental group of dieters (*n* = 107; 67.50% female) received feedback on their GAS scores, but the control group did not (*n* = 111; 62.30% female). Compared with the control group, the experimental group reported a greater need for information, greater readiness to change, and higher perceived situational confidence in resisting food that was inconsistent with their dieting goals. To conclude, the GAS could be used in health settings to provide clients and providers with an objective, fast measure of commitment to achieving health-related goals. Moreover, immediate feedback on health-related goals may improve change motivation.

## 1. Introduction

In everyday life, humans strive to manage and pursue multiple goals simultaneously [1,2,3,4]. There is a common consensus that people’s goals, as desired endpoints to be achieved, and the ways they strive for them are linked to their success and well-being [5,6,7,8,9].

Goals constitute a central construct of human motivation [7,10,11,12]. Whereas a goal represents a *desired* state that can guide action [7], motivation comprises the psychological processes involved in the arousal, direction, intensity, and persistence of activities that are goal related [5,6,7]. Motivation starts as soon as people become committed to pursuing a goal. According to the motivational model [5,6], factors within people can affect their chances of achieving their desired goals. These factors can be classified into three broad properties of motivation: (a) goal *significance* or the extent to which people want to achieve a given goal. The significance can be measured on a continuum of liking–disliking or by having plans to accomplish the goal in the foreseeable future; (b) goal interrelationships, or how much a goal facilitates or impedes achieving other desired, significant goals that would satisfy them; and (c) goal *value*, or people’s expected net emotional payoff from achieving the goal, taking into account the difficulties they might encounter when achieving the goal. Expected net happiness from achieving a goal or expected sadness if the goal cannot be achieved determines how salient the goal is in the person’s life.

Because a goal is a desired endpoint in the future that a person wants to achieve to change a less desirable state, people perceive or visualize an emotional payoff (greater happiness or less discomfort) while striving to achieve the goal. This means that individuals may need to disengage themselves from other goals or set new goals as prerequisites of striving for the desired goal; competing goals may create incompatible responses [5,13,14,15] that can interfere with the pursuit of the most desired goal, and they might cause mixed emotions [1,16,17] that may directly or indirectly decrease the person’s satisfaction with life [17,18]. However, there is also evidence that inter-goal conflict or facilitation may not affect goal pursuit merely because it influences people’s mood [16]. The widely used Inter-Goal Relations Questionnaire (RQ) [19,20] in health-related settings evaluates the extent to which pursuing a goal contradicts other goals or imposes time, energy, and financial constraints on pursuing other goals or the extent to which it sets the stage for and facilitates achieving other goals. Therefore, the IRQ evaluates inter-goal relationships mainly through their instrumentality dimension. Research exploring the impact of inter-goal facilitation/conflict in the pursuit of health-related goals has led to mixed results. Riediger and Freund [20] reported that intergoal facilitation is related to the intensity of physical exercise, regardless of the age of the participants. Elliston et al. [16], administering the IRQ, reported that *inter*-goal conflict and facilitation cannot predict dieters’ weight loss in the long term via changes in their mood or snacking and food consumption. It seems that the strength of a goal system comprising conflicting and facilitating goals depends on essential factors such as expectancy, value, opportunities, costs, and people’s goal-seeking characteristics, which determine their ways of dealing with their goals [21]. For unrelenting pursuit, a desired, new goal should theoretically be higher in the three properties of motivation (i.e., importance, instrumentality, and value) than a less desirable, habitual goal. However, the perceived distance from the current situation to the new, more desirable situation can be far enough to dampen the person’s motivation to pursue the new goal [13,22,23,24,25]. However, perseverance in pursuing a distant goal can be buttressed by exercising cognitive control to maintain the importance and value of the goal [26,27].

To summarize, individuals’ desire to become committed to pursuing a new goal to change their current situation might be affected by their degree of satisfaction with their current status. That is, some people state that (a) they know that a particular change is crucial for them to achieve; (b) they will enjoy achieving their goal; and (c) they think achieving it will help them to other goals in their lives better. They may, nevertheless, feel comfortable with their current situation and still think that (a) the new goal is not important enough for them to actively pursue it; (b) their life is enjoyable even without striving for the new goal; and (c) they can still achieve their other goals without committing themselves to achieve a new desirable goal.

The goal of the present study was to investigate how becoming committed to pursuing a new goal can be a way to resolve ambivalence such as that described above. We also aimed to answer whether congruence or incongruence between the *desire* to achieve a particular goal and the *reluctance* to achieve it at the same time can predict the likelihood of one’s becoming committed to the pursuit of the goal. We, therefore, developed a scale specifically for measuring people’s certainty or uncertainty about the three motivational indices described above. The rationale of the scale was based on addressing *intra*-goal conflict or goal ambivalence. We investigated these questions in a sample of dieters. We report how we determined our sample size, all data exclusions, all manipulations, and all measures in the study.

## 2. Study 1

### 2.1. Method

#### 2.1.1. Participants

The participants (*n* = 334; mean age = 35 years; *SD* = 12.00; 74.6% female) were individuals referred to a private dieting clinic for management of their weight either for non-medical reasons or for a medical condition (47%) (e.g., diabetes, breast-feeding) that could affect their diet. On the Goal Ambivalence Scale (GAS; see below), 32 participants (9.6%) specified that they did not have any health-related concerns; hence, they did not complete the GAS. More than half of the participants (*n* = 142; 65%) were following a diet at the time they completed the study measures. They varied from those who were dieting for the first time (46%) to those who were resuming their diet for a second time or more (18%).

#### 2.1.2. Goal Ambivalence Scale

We developed the Goal Ambivalence Scale (GAS; Appendix A) based on components of goal commitment based on Cox and Klinger’s motivational model [5,6,28] and the transtheoretical model of health-related behavior [29]. We focused on three essential components of motivation that affect goal commitment—goal importance, instrumentality, and value. Goal importance is the strength of the person’s desire to reach the goal. Goal instrumentality is the extent to which achieving a given goal can facilitate achieving other goals in the person’s life, and the value of a goal refers to the net emotional payoff that one expects from achieving the goal. The GAS, therefore, consists of two parts. Part A measures the person’s agreement with the three indices—wanting, instrumentality, and expected emotional payoff if one achieves a particular goal. Therefore, we believe that Part A represents the *Affirmation* of one’s commitment to strive for the desired change. Part B measures the person’s agreement with each of the same three indices if the person does *not* achieve the same goal. Therefore, we believe that high scores in Part B represent the *Refutation* of one’s commitment to strive for the desired change.

Respondents are asked to mark their agreement with each item out of four choice response options. Scores can range between *zero* for the least agreement with an item and *three* for the maximum agreement with an item. The Goal Commitment Likelihood Score (GCLS) was calculated as Part A total score *minus* Part B total score. Because the participants were recruited from a dieting clinic, the questionnaire also asked whether they had other health-related goals or desires. In Study 1, the instructions on calculating the GCLS and its interpretation were omitted from the bottom of the scale (Appendix A).

#### 2.1.3. Health-Related Concerns and Actions (HRCA)

An author-compiled questionnaire (called Health-Related Concerns and Actions; see Appendix B) was developed to measure three aspects of respondents’ motivation: (a) their health-related concerns; (b) their desire to do something about their health-related concerns; and (c) their desire to receive relevant information to help them resolve each of their health-related concerns. They were also asked to provide contact details if they wanted to participate in a subsequent similar study. The number of other health-related concerns marked on the questionnaire served as a covariate in the regression model that predicted the study’s dependent variables, i.e., the participants’ desire to *do* something about their diet and to receive informational tips from the researchers (in addition to visiting a specialist doctor) related to their GAS scores.

### 2.2. Procedure

The study was announced on the dieting clinic’s notice board as a study that assesses people’s attitudes about their health status and their diet. Informed consent was obtained from participants who volunteered for the study. Upon giving their consent, they were provided with a series of study questionnaires to complete while waiting to see their doctor. They could resume completing the questionnaires after seeing their doctor if they had not already completed them. The Ethics Committee of Ferdowsi University of Mashhad, which is regulated by the Iranian National Committee for Ethics in Biomedical Research, approved the study protocol and the materials that the participants completed (IR.UM.REC.1400.055).

Qualitative and quantitative evaluation of the GAS content was obtained by collecting data from five specialists in clinical and health psychology, dieticians, specialists in the treatment of diabetes, and substance abuse. The specialists evaluated the scale items in terms of their grammar, wording, item allocation, and scoring method. Next, we conducted a pilot study with a few samples of dieters (*n* = 10 & 10) and people with diabetes (*n* = 10) in clinical settings similar to the main study to ascertain the clarity and ease of understating the GAS items and instructions. Adjustments to the GAS were made based on the feedback collected from the experts and the patients. Again, the same experts were consulted about the GAS items and instructions as follows.

### 2.3. Results

The Content Validity Ratio (CVR) and the Content Validity Index (CVI) were used to assess the quantitative content of each item [30]. The CVR of each item was determined by considering its relevancy, clarity, and simplicity. The specialists rated these items on a three-point scale: 1 = necessary, 2 = useful but not necessary, and 3 = not necessary. The mean judgments of the experts [31] were used to calculate the CVR scores, which, for all GAS items, were greater than 0.99, i.e., the minimum acceptable value. For the CVI index, the specialists determined the relevance of each item on a four-point Likert scale (1 = not relevant, 2 = somehow relevant, 3 = relevant, and 4 = completely relevant). The values for all the GAS items were larger than CVI > 0.79; hence, all items were retained [30,31].

We also conducted an Exploratory Factor Analysis (EFA) using Principal Axis Factoring (PAF). A Kaiser–Meyer–Olkin measure of sampling adequacy of 0.76 and a Bartlett’s test of sphericity of 518.44 (*p* = 0.001) supported the model’s fitness. An initial solution consisting of two components with eigenvalues greater than 1.0 was extracted. The two factors accounted for 52.30% of the variance. An additional EFA with equamax rotation and Kaiser normalization was calculated, which resulted in two factors with loadings that clearly suggest that the two parts of the GAS clustered together on each respective factor (Table 1). Based on the theory and rationale of the scale, we named the first factor *Affirmation* (equal to items in Part A) and the second factor *Refutation* (equal to items in Part B).

Moreover, to evaluate the factor structure of the GAS, we calculated a Confirmatory Factor Analysis (CFA) that employed structural equation modeling using AMOS. The maximum likelihood estimator (robust) was used for the CFA. Indices of model fit were calculated. The value χ^2^ = 16.13 (5) was significant (*p* = 0.006), but, as Byrne [23] stated in the case of computing CFAs on self-report questionnaires, it is unexpected to get a nonsignificant χ^2^ because it is very sensitive to sample size. However, other fit indices, including NFI = 0.96 and CFI = 0.98, and an RMSEA of 0.087 with a nonsignificant PCLOSE value of 0.081, suggested that the model had an acceptable fit. Moreover, a small value of the ECVI = 0.20 was also calculated, representing a greater possibility for model replication in future samples [32]. Figure 1 shows the CFA model loadings for the items in the two latent factors which represent Affirmation (Part A) and Refutation (Part B). The negative covariance of 0.52 between the two factors was in accordance with the theory and the rationale of the scale. The item loadings were greater than 0.50 as per the recommended value for a newly developed scale—for established scales, they should be greater than 0.06 with an R2 greater than 0.40; otherwise, they can be considered for deletion from the model if the inclusion of the items is not supported by theory and evidence [33]. To conclude, the CFA results in the first study supported the scale’s factor structure based on the initial EFA reported earlier.

Next, to explore the dimensionality of the GAS parts and to test the correlation of each dimension with scores from each part and the scale’s GCLS (Table 2 and Table 3), the GAS data were subjected to a principal component analysis (PCA). A Kaiser–Meyer–Olkin measure of sampling adequacy of 0.76 and a Bartlett’s test of sphericity of 520.44 (*p* = 0.001) supported the fitness of the model. An initial solution consisting of two components with eigenvalues greater than 1.0 was extracted, which, together, accounted for 67.73% of the variance. An additional PCA with varimax rotation and Kaiser normalization was calculated, which resulted in two components with loadings that clearly suggest that the two parts of the GAS clustered together on each respective component (Table 1). An intercorrelation matrix of the two components and GAS Part A and Part B showed that Component 1 (representing Affirmation) was strongly correlated with GAS, Part A, and Component 2 (representing Refutation) was strongly associated with GAS, Part B (Table 2).

To calculate the reliability of the GAS, items 3–6 were first reverse scored. Total scores were calculated for each part of the scale and were then summed. Next, the GAS item scores from 300 participants were subjected to a reliability analysis using internal consistency, which yielded a Cronbach’s alpha of α = 0.78. There was no item whose deletion improved the reliability of the scale. The results of an analysis of variance using Tukey’s test for nonadditivity were significant for both between items (*F*_(5, 299)_ = 27.60; *p* < 0.001) and nonadditivity (*F*_(1, 299)_ = 10.37; *p* < 0.001). The results of Hotelling’s *t*-squared test = 77.59 (*F*_(5, 295)_ = 15.31; *p* < 0.0001) also supported the adequacy of the data analysis. Reliability statistics were also calculated using a split test, which yielded a Cronbach’s alpha of α = 0.78 for Part A (three items) and α = 0.72 for Part B (three items); Spearman-Brown coefficient = 0.59; and Guttman split-half coefficient = 0.58.

To examine the predictive validity of the GAS, we conducted a hierarchical regression analysis. In the model, *frequency of dieting during the last year* was entered as the dependent variable; sex and age were entered in the first step of the model to control for their effects; and Component 1 and Component 2 were entered as the predictor variables in the second step. There was a significant change in the model’s variance that was accounted for by Component 1 in the second step (ΔR^2^ = 0.036; *F*_(2, 290)_ = 4.60, *p* = 0.011; *t* = 3.02, *p* = 0.003) after the effects for sex and age had been controlled in the first step (*p* < 0.05). A second regression analysis was conducted with the same variables, except for the second step, in which the GAS Part A and Part B total scores were entered. Again, the GAS Part A total score was the only significant predictor variable (ΔR^2^ = 0.038; *F*_(2, 292)_ = 4.98, *p* = 0.007; *t* = 3.12, *p* = 0.002) beyond the portion of the variance that had already been explained by the variables in the first step (*p* < 0.05). In the third similar model, the Goal Commitment Likelihood Scores (GCLSs) (calculated as Part A *minus* Part B) were entered in the second step. The results showed that the GCL scores predicted a significant change in the variance of the model over the portion already explained in the first step (ΔR^2^ = 0.018; *F*_(2, 294)_ = 3.92, *p* = 0.049; *t* = 1.98, *p* = 0.049). Finally, we conducted a hierarchical regression analysis in which the number of informational tips that respondents wished to receive was entered as the dependent variable; sex, age, dieting frequency during the last year, and total health-related concerns (problems) were entered in the first step of the model to control for their effects; and GCL scores were entered as the predictor variable in the second step. There was a significant change in the explained variance because of the variables entered in the first step (*F*_(4, 285)_ = 5.17, *p* = 0.001). In the first step, sex (*t* = 3.25; *p* = 0.001) and total problems (*t* = 2.50; *p* = 0.013) were significant predictors of the need for more informational tips. The GCL scores entered in the second step also significantly added to the model’s variance (ΔR^2^ = 0.021; *F*_(1, 285)_ = 5.50, *p* = 0.012; *t* = 2.53, *p* = 0.012) after the variables entered in the first step had been controlled (*p* < 0.05).

## 3. Discussion

The results of Study 1 suggest that the GAS is a reliable and valid measure of commitment to a health-related goal. The GAS yielded the hypothesized factor loadings for Part A and Part B of the measure. It is noteworthy that the goal commitment scores predicted the participants’ need to receive further information and assistance regardless of their gender and health-related problems. Therefore, it seems that the GAS can provide patients/clients and healthcare providers with a quick evaluation of clients’ commitment to a particular health-related concern or goal that they want to pursue.

In the subsequent study, we investigated the immediate effect of providing patients with an estimate of their commitment on the GAS. We hypothesized that providing immediate feedback would improve participants’ goal-related motivation. Goal-related motivation was defined as (a) the participants’ self-reported willingness to receive further informational tips on health-related issues and (b) their confidence in being able to adhere to their treatment goals and resist their habitual, unhealthy behaviors in various tempting situations as measured by the Situational Confidence Questionnaire. Goal-related effort was defined as the participants’ self-reported inclination to take a list of health-related actions to improve their health condition, as reflected in their score on a modified version of the Readiness to Change Questionnaire. Study 2 describes the use of the GAS at a dieting clinic to predict respondents’ need for more information, their readiness to change, and their perceived situational confidence in resisting food that is inconsistent with their dieting goals.

## 4. Study 2

### 4.1. Participants

Participants (*n* = 218; mean age = 32.7 years; *SD* = 12.85; 63.75% female; mean education = 14.5 years; *SD* = 3.52) were patients receiving therapeutic services at a private dieting clinic. They were randomly assigned to an experimental group that received the CGS (*n* = 107) or to a control group that did not receive the CGS (*n* = 111). However, on the GAS, 15 participants (14.8%) in the experimental group indicated that they did not have a health-related concern; hence, they did not complete the GAS. Therefore, 92 participants (mean age = 32.78 years; *SD* = 14.16; 75.00% female; mean education = 14.09 years, *SD* = 3.13) were retained in the experimental group for the subsequent analyses. Only six participants in the control group indicated that they had no health-related concerns; hence, they were not included in the analyses to test the hypotheses (*n* = 104; mean age = 32.73 years; *SD* = 12.52; 73.10% female; mean education = 14.35 years, *SD* = 3.28); the resulting total *n* was 196. Table 4 compares the characteristics of the two samples.

### 4.2. Instruments

The instruments administered included the demographic and general information questionnaire, Goal Ambivalence Scale (as in Study 1, see Appendix A), the HRCA questionnaire (as in Study 1; see Appendix B), and modified versions of the Situational Confidence Questionnaire and Readiness to Change Questionnaire.

A modified version of the Situational Confidence Questionnaire (SCQ) [34] was used to measure drug abusers’ confidence in their ability to resist drugs in eight situations (viz., those in which they experience positive emotions, negative emotions, temptation and urges, positive social situations, social tensions, social problems at work, physical discomfort, and the testing of self-control). The SCQ was modified to measure the participant’s ability to resist forbidden food and drinks. Instructions on the measure were modified, and the stems of the items were changed to “I would be able to resist the urge to drink and eat…”. For the Persian food version of the SCQ, a Cronbach’s α of 0.93 was previously calculated.

The Readiness to Change Questionnaire (RTCQ) [35] was originally developed to measure motivation for change among alcohol and drug users. The questionnaire assigns each respondent to one of the three stages of change: precontemplation, contemplation, or action. The scores on each scale can range from −8 to +8, with negative scores reflecting an overall disagreement with the items, whereas positive scores reflect overall agreement with items in the scale. The items on the RTCQ were modified to measure the ability to resist unhealthy, high-calorie foods and sugary drinks. Therefore, modifying the items entailed a simple wording change from alcohol to food and beverages. We calculated a Cronbach’s α of 0.70 for the Persian version of the food RTCQ that was used in the present study.

### 4.3. Procedure

The study was announced to the patients in the same way as the first study. Again, informed consent was obtained from all participants who volunteered to take part in the study. Two sets of envelopes were prepared, which contained the study instruments. One set of envelopes contained the GAS and was given to the experimental group; the other set did not contain the GAS and was given to the control group. The envelopes from each set were randomly given to the volunteering patients. The GAS that was given to the experimental group contained simple instructions at the bottom of the page on how to calculate a total scale (i.e., the sum of Part A minus the sum of Part B). In this way, they were able to see an interpretation of their total score at the bottom of the scale (Appendix A). The interpretation of the range for each score was based on the authors’ familiarity with the core concept underlying the questions and their clinical experience.

After completing the demographic questionnaire and three questions on their dieting history, current status, and reasons for dieting, the experimental group was instructed to proceed with completing the CGS if they had a health-related concern (if they did not, they could skip it, and responses for them were not included in the data analysis). The control group was not introduced to the GAS, and they were not required to provide information on whether they had a health-related concern. With these exemptions, participants answered the same battery of questionnaires, i.e., the SCQ, RTC, and the Health-Related Concerns and Actions questionnaire. At the end of the testing session, they were thanked for their participation.

### 4.4. Results

#### 4.4.1. Primary Analyses

Four sets of analyses were performed. *First*, a series of nonparametric tests (i.e., Mann–Whitney U and Kolmogorov–Smirnov tests) were conducted to compare the two groups on the distribution of sex (male, female), marital status (single, living together), medical conditions affecting their eating (yes, no), and lost vs. gained weight during dieting; none of the nonparametric tests was significant (*p* > 0.05). *Second*, the results of three independent *t*-tests showed that the means for age, dieting history (i.e., the number of previous dieting attempts), and the number of self-reported health-related concerns were not different for the two groups (*p* > 0.05). *Third*, a correlation matrix was calculated for the measures that were expected to covariate with the main outcome measures. The pattern of significant correlations was slightly different for females vs. males (Table 5). *Fourth*, a series of univariate (ANCOVAs) and multivariate analyses of covariance (MANCOVAs) were conducted to test the study hypotheses (Table 6). For all ANCOVA models, the results of Box’s M test of equality of covariance matrices and Levene’s test of equality of error variances confirmed the fitness of the data for each model.

Based on the result of the correlation matrix, the first series of ANCOVAs (Models 1–4) was conducted to test the hypothesis that providing fast, brief feedback on the dieters’ health-related goal commitment using their results from the CGS would increase their motivation for successful dieting. In the ANCOVA models, the dependent variables (DVs) were (a) the number of health information tips the dieters indicated they would like to receive (H info; Model 1); (b) the number of actions they specified that they would take soon to improve their health (H actions; Model 2); (c) their Readiness to Change (RTC) total score (Model 3); and (d) their Situational Confidence Questionnaire (SCQ) total score (Model 4). Group (experimental vs. control) and sex and their interaction term were entered as the predictors (independent variables; IVs); and age, education (in years), number of health-related concerns, and dieting history were entered as covariates. In the first MANCOVA (Model 5), the DVs were the RTC subscale scores (i.e., precontemplation, contemplation, and action), whereas the IVs and the covariates were the same as in the first model. In the second MANCOVA (Model 6), the DVs were the SCQ subscale scores (i.e., on pleasant emotions, unpleasant emotions, temptation and urges, positive social situations, social tensions, social problems at work, physical discomfort, and the testing of self-control). All the other variables in the model were the same as in the previous models. Table 6 shows the results of the four ANCOVA and two MANCOVA models.

As Table 6 shows, in the four ANCOVA models, there were significant main effects for Group for Models 2–4, testing (a) the need for more information, (b) Readiness to Change total score, and (c) Situational Confidence total score, respectively; however, in none of the models was Sex or its interaction with Group significant. Education and number of health-related concerns (No. HRC) were significant covariates in Model 2, but No. HRC was the only significant covariate in Model 3. The experimental group had higher scores than the control group in their need for more information, RTC total score, and SCQ total score. The difference between the two groups was not significant (*p* = 0.53) on action plans. In the fifth model (MANCOVA), there was a significant main effect for Group but not for Sex or the Sex × Group interaction. Age, dieting history, and the number of health-related concerns were significant covariates in the model. Participants in the control group had higher scores than the experimental group on the RTC precontemplation subscale. In the sixth model (MANCOVA), the main effect for Group approached significance (*p* = 0.064). However, the experimental group was higher than the control group in self-estimations of their ability to resist eating forbidden foods and drinks while experiencing pleasant or unpleasant emotions and while experiencing urges and temptations to consume forbidden foods and beverages because of experiencing social problems at work. 

#### 4.4.2. Secondary Analyses

First, similar to the first study, we repeated the EFA (using the PAF) and PCA analyses on the GAS data collected from the experimental group in the second study, the results of which are shown in Table 1. Both models met the fit indices and resulted in similar factors/components to those observed in the first study. In the PAF model, the first factor explained 31.25% of the variance, and the second factor explained 23.12% of the variance for a total of 54.36. In the PCA model, the first component explained 37.30% of the variance, and the second component explained 29.66% of the variance for a total of 66.96%. Likewise, we calculated another CFA analysis on the GAS data for participants in the experimental group. The indices of model fit were χ^2^ = 8.83 (5), which was nonsignificant (*p* = 0.11), NFI = 0.95, CFI = 0.98, and an RMSEA of 0.09 with a nonsignificant PCLOSE value of 0.20, suggesting that the model had an acceptable fit. Figure 2 shows the result of the CFA model factor loadings, which replicates the two latent factors from Study 1, representing Affirmation (Part A) and Refutation (Part B). Again, the item loadings were greater than 0.50, the recommended value for a newly developed scale [34], and there was a negative covariance between the two factors. The negative covariance supported the pattern of correlation between the scale’s GCLS and the PCA Component 1, representing Affirmation, r = 0.59, and the PCA Component 2, representing Refutation, r = −0.74. To conclude, it seems that the results of CFA further support the GAS.

Second, the following complementary analyses were conducted with participants in both groups. To investigate the effect of medical necessities for dieting on the GAS intervention, parallel models, similar to the ANCOVA and MANCOVA models reported above, except for changes in the predictor variables, were conducted. In all the models, the IVs were Group, Medical Conditions Affecting Eating Behavior (MCAEs; present vs. absent), and their interaction. Of the 92 participants in the experimental group who answered the question, 30 stated that they had a medical reason for being referred to the diet clinic, and of the 103 participants in the control group who answered the same question, 57 had a medical condition. For the parallel ANCOVA Models 1–4, the only significant main effect was Group (*F*_(1, 188)_ = 4.77; *p* = 0.030, η2 = 0.025) in the model testing the SCA total score; neither MCAE nor the Group × MCAE interaction was significant (*p* > 0.05). In the model, age was a significant covariate. For parallel MANCOVA Model 5, there was no significant main effect for Group, MCAD, or the Group × MCAD interaction, but age, education, and the number of health concerns were significant covariates only in Model 5. Similarly, for parallel Model 6, there was no significant main effect for Group, MCAD, or the Group × MCAD interaction, and none of the covariates was significant. In similar models, the three-way Group × MCAE × Sex interactions were tested, but none of the models with the three-way interaction terms was significant.

Fourth, a correlation matrix was calculated for the GAS PCA Component 1 (Affirmation) and Component 2 (Refutation) and Goal Commitment Likelihood Score (GCLS) with the study outcome measures for the experimental group—i.e., the group that completed the GAS. The significant correlations were as hypothesized for the GAS scores and the RTC subscales and total score; however, there was no significant correlation between the SCQ subscales and total score.

## 5. Discussion

Commitment is an essential component of one’s motivation to pursue a goal. People’s degree of commitment affects their decision to initiate goal pursuit and exercise sufficient perseverance to achieve the desired endpoint. In health-related contexts, readiness to change an unhealthy behavior means moving through successive stages of change (from precontemplation to contemplation to action) to achieve the desired health state [36,37]. Commitment, however, must still be mustered to maintain the change and prevent relapse. This requires a state of mind that can resist tempting situations that might trigger undesirable older habits [38,39]. When people set their mind to achieving a goal, a dynamic mental state is instigated, which is known as a *current concern* [40]. This motivational force continues for the duration of each goal pursuit between two points in time, the point when one first becomes committed to pursuing the goal and the time when the goal is achieved or the person becomes disengaged from pursuing it.

Various factors affect one’s degree of commitment to achieving a goal. These include the importance of the goal, the instrumentality of the goal in facilitating or interfering with the achievement of other goals, and the expected emotional payoff if one achieves the goal [40,41]. In addition, people might experience unresolved ambivalence, making them unsure whether they want to strive to achieve the goal. This is because acquiring the goal might intensify the individual’s emotional satisfaction but also increase negative emotions, such as anxiety or depression. The person must decide, therefore, whether it is worthwhile to pursue the goal considering the number of positive and negative outcomes that could ensue and whether achieving the goal would genuinely enable them to feel better.

We developed the Goal Ambivalence Scale (GAS) to quantity people’s commitment to a health-related goal based on self-reported ratings of the three most important determinants of goal commitment. The results of the first study supported the reliability and validity of the GAS, suggesting that it might have clinical utility by providing immediate feedback to patients and their healthcare providers. The results of the second study supported our hypothesis that a brief self-assessment of patients’ commitment to their health-related goals using the GAS can improve their willingness to seek more information that might assist them in taking more action to achieve their health-related goals. The improvement was maintained even after the effects of age, education, dieting history, and the number of health-related concerns had been controlled. Additionally, participants who were provided with an interpretation of their scores on the GAS reported even greater readiness to change and more confidence in their ability to resist tempting situations that could jeopardize their dieting goal. The feedback might have encouraged them to think more seriously about their reasons for visiting the dieting clinic and helped them to see, in more concrete ways, the elements comprising their decision to try to control their weight. Prior evidence shows that providing people with contingent feedback on their performance can improve their sense of control [9,42].

The study was conducted with a sample of volunteers in a dieting clinic. Therefore, the generalization of the findings to other health-related conditions warrants further investigation. Decisions to change health-related behaviors might range from, for example, deciding to drink more water, apply sun cream in sunny weather, exercise more vigorously, or quit smoking to more serious decisions, such as recovering from addiction or adhering to the treatment for cancer. It should also be noted that members of different cultures might respond in different ways to the GAS and the feedback it provides about their commitment to achieving their health-related goals. Moreover, the control group did not receive any form of feedback; therefore, observed changes could be attributable to the feedback they received. Future studies could use sham feedback with the control sample to control for this potential issue.

Regarding future research, we suggest that the benefits of using the GAS with patients should be evaluated among treatment staff, such as physicians, nurses, counselors, and therapists, and with nonclinical samples in nonclinical settings. We also suggest that an evaluation study should be conducted in which treatment staff discuss with patients their GAS scores to help them intensify their commitment to achieving their health-related goals. Additionally, administering competing measures of goal commitment would help further explore the predictive validity of the GAS relative to other different established scales of health intention. It seems necessary that future studies explore the long-term outcomes of providing patients with their GCLS in clinical and nonclinical settings.

To conclude, the Goal Ambivalence Scale is a brief measure of one’s commitment to achieving health-related goals. The scale can be used with patients in health-related settings to provide them with simple and immediate feedback on their commitment to achieving health-related goals.

## Figures and Tables

**Figure 1 behavsci-12-00441-f001:**
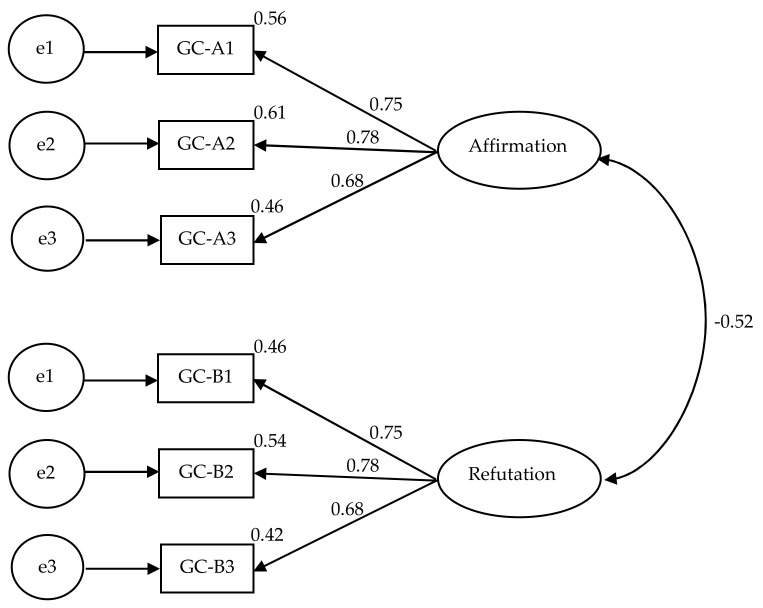
Results of confirmatory factor analysis showing items’ loadings for a two-factor model for Goal Ambivalence Scale based on data from Study 1.

**Figure 2 behavsci-12-00441-f002:**
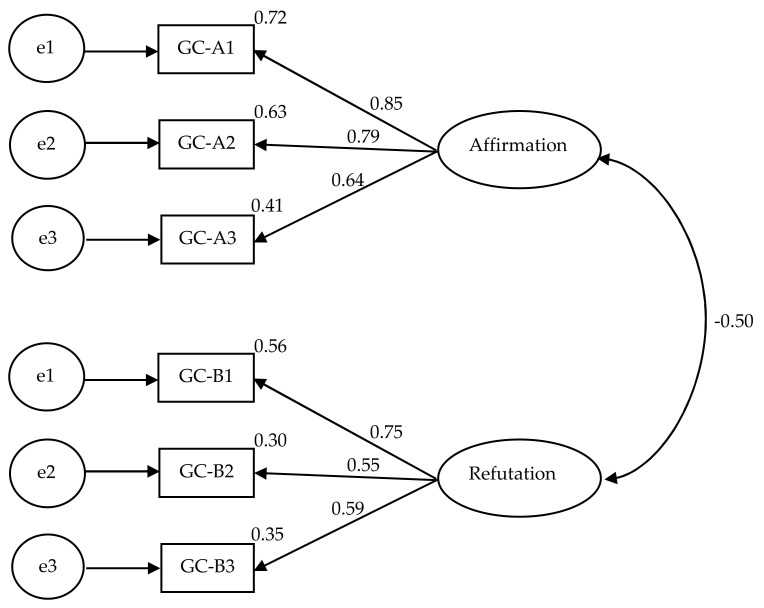
Results of confirmatory factor analysis showing items’ loadings for a two-factor model for Goal Ambivalence Scale based on data from Study 2.

**Table 1 behavsci-12-00441-t001:** Rotated factors and component matrices using principal axis factoring and principal component analysis for the Goal Ambivalence Scale (GAS) items.

	Study 1				Study 2			
GAS Items	Principal Axis Factoring		Principal Component Analysis		Principal Axis Factoring		Principal Component Analysis	
	1	2	1	2	1	2	1	2
**Part A (affirmation)**								
Item 1	0.78		0.84		0.92		0.92	
Item 2	0.73		0.82		0.80		0.87	
Item 3	0.64		0.79		0.52		0.68	
**Part B (refutation)**								
Item 1		0.56		0.68		0.47		0.63
Item 2		0.78		0.84		0.47		0.78
Item 3		0.63		0.82		0.87		0.81

Note. PAF and PCA: equamax rotation converged in three iterations.

**Table 2 behavsci-12-00441-t002:** Correlation matrix for GAS PCA components and total scores for Part A and B and the scale’s total score.

	Component 1	Component 2	GAS Part A (Affirmation)	GAS Part B (Refutation)
GAS Part A	0.98 **	0.20 **		
GAS Part B	0.23 **	0.97 **	0.40 **	
GCLS	0.69 **	0.72 **	0.83 **	0.86 **

Note. GCLS = Goal Commitment Likelihood Score. ** Correlation is significant at the 0.01 level (one-tailed).

**Table 3 behavsci-12-00441-t003:** Correlation matrix of GAS scale PCA Component 1 (Affirmation) and Component 2 (Refutation), Goal Commitment Likelihood Score (GCLS) with the study outcome measures for the experimental group.

	PCA Affirmation	PCA Refutation	GCLS
Refutation	0.001		
GCLS	0.59 **	−0.74 **	
Age	0.088	−0.25 *	0.22 *
Education	−0.013	0.15	−0.14
Diet History	0.18 *	−0.035	0.13
NO. health con.	0.19 *	0.082	0.069
Actions to be taken	−0.020	−0.11	0.088
Health info. Required	0.22 *	0.016	0.16
RTC Precontemplation	−0.19 *	0.14	−0.21 *
RTC Contemplation	0.086	−0.38 **	0.36 **
RTC Action	0.11	−0.35 **	0.39 **
RTC total score	0.23 *	−0.43 **	0.51 **
SCQ: Pleasant Emotions	0.043	−0.042	0.077
SCQ: Unpleasant Emotions	−0.13	0.050	−0.097
SCQ: Urges and Temptations	0.048	−0.005	0.029
SCQ: Positive Social Situation	0.016	−0.012	0.013
SCQ: Social Tension	−0.022	0.073	−0.070
SCQ: Social problems at work	−0.081	0.056	−0.060
SCQ: Testing Personal Control	−0.018	−0.126	0.093
SCQ: Physical Discomfort	−0.090	−0.079	−0.005
SCQ total score	−0.055	0.004	−0.029

Note. NO. health con. = Number of health-related concerns; Health info. Required = Health Information Required; RTC = Readiness to Change; SCQ = Situational Confidence Questionnaire. ** *p* < 0.01; * *p* < 0.05.

**Table 4 behavsci-12-00441-t004:** Comparison of group characteristics in the second study.

	Experimental Group		Control Group			
	Mean	*SD*	Mean	*SD*	*t* (194)	*p*
Age	32.78	14.16	32.68	10.94	0.056	0.96
Education	14.09	3.14	14.59	3.41	−1.062	0.28
Diet history	1.84	1.29	2.13	1.47	−1.50	0.14
No. of health concerns	3.30	2.15	3.00	1.78	1.047	0.30
Sum of actions	1.29	0.71925	1.2816	.63	0.12	0.90
Need for information	1.55	1.918	1.16	1.41	1.64	0.10
RTC: Precontemplation	−1.95	3.72	−1.067	3.28	−1.75	0.083
RTC: Contemplation	4.12	3.33	3.63	3.47	0.99	0.32
RTC: Action	3.13	3.46	2.75	3.29	0.77	0.44
RTC total score	9.17	6.65	7.59	6.84	1.63	0.10
SCQ: Pleasant emotions	75.11	20.94	69.23	21.43	1.94	0.054
SCQ: Unpleasant emotions	66.95	57.11	60.00	26.84	1.11	0.26
SCQ: Urges and temptations	62.88	26.36	53.89	24.95	2.45	0.015
SCQ: Positive social situation	64.67	25.74	56.15	24.93	2.35	0.020
SCQ: Social tension	63.91	30.91	59.13	28.18	1.13	0.26
SCQ: Social problems at work	73.040	56.22	57.40	27.76	2.51	0.013
SCQ: Testing personal control	64.13	26.024	60.57	24.36	0.99	0.32
SCQ: Physical discomfort	65.87	27.46	61.051	26.17	1.26	0.21
SCQ total score	67.029	23.99	59.65	18.77	2.41	0.017

Note. No. of health concerns = number of health-related concerns.

**Table 5 behavsci-12-00441-t005:** Correlation matrix of age, education, dieting history, number of health-related concerns, total actions to be taken, further information required, Readiness to Change total score, and Situational Confidence Questionnaire total score, separately for females and males.

	Age	Edu.	Diet History	No. Health Con.	Sum of Actions	Health Info. Required	RTC Total Score	SCQ Total Score
Age		0.46 **	0.058	−0.035	0.19	0.31 *	−0.12	0.38 **
Edu.	−0.17 *		0.15	0.15	−0.039	0.27	0.025	0.33 *
Diet history	0.16 *	0.12		0.18	−0.15	0.14	0.31 *	−0.064
No. health con.	0.031	−0.016	0.20 *		0.16	0.24	0.25	0.043
Sum of actions	0.001	−0.064	−0.069	0.20 *		−0.096	−0.031	0.13
Health info. required	0.005	0.13	−0.022	0.26 **	0.21 *		0.18	0.24
RTC total score	0.22 **	0.049	0.28 **	0.22 **	0.011	0.11		0.07
SCQ total score	0.083	−0.034	−0.11	−0.16 *	−0.14	0.017	−0.062	

Note. Correlations below the diagonal line are for females; above the diagonal line are for males. Edu. = education in years; No. health con. = number of health-related concerns; Health info. required = health information required; RTC = Readiness to Change; SCQ = Situational Confidence Questionnaire. ** *p* < 0.01; * *p* < 0.05.

**Table 6 behavsci-12-00441-t006:** Results of three MANCOVAs testing to evaluate the main hypothesis about the immediate effects of a brief goal commitment feedback on dieters’ need for information, plans for taking action, Readiness to Change (RTC), Situational Confidence (SC), and the RTC and SCQ subscales.

	Main Effect		Interaction	Covariate				Pairwise Comparison		
Model	Group F [p] η2	Sex F [p] η2	Group*Sex F [p] η2	Age F [p] η2	Diet Hist. F [p] η2	Edu. F [p] η2	No. Health Concern F [p] η2	Exp *M* (*SD*)	Ctrl *M* (*SD*)	Result
** Models 1–4, F (1, 188) **										
1. Plan for action	0.39 [0.53] 0.002	1.50 [0.065] 0.018	0.40 [0.53] 0.002	0.33 [0.57] 0.002	2.80 [0.09] 0.015	0.50 [0.48] 0.003	9.15 [0.003] 0.047	1.29 (0.71)	1.25 (0.60)	
2. Need for info.	4.16 [0.043] 0.022	1.41 [0.221] 0.008	2.029 [0.156] 0.011	1.34 [0.24] 0.007	0.88 [0.35] 0.005	6.19 [0.014] 0.032	13.49 [0.001] 0.068	1.55 (1.92)	1.16 (1.42)	
3. RTC total score	5.06 [0.026] 0.026	0.28 [0.59] 0.001	1.22 [0.27] 0.006	2.69 [0.102] 0.014	11.03 [0.001] 0.055	0.31 [0.57] 0.002	3.59 [0.026] 0.026	9.17 (6.65)	7.27 (7.07)	Exp > Ctrl
4. SCQ total score	5.18 [0.017] 0.030	0.40 [0.52] 0.002	0.21 [0.64] 0.001	5.24 [0.023] 0.027	1.65 [0.20] 0.009	2.022 [0.15] 0.011	2.76 [0.098] 0.014	67.02 (23.99)	60.21 (19.00)	Exp > Ctrl
** Model 5, F (3, 190) **	2.65 [0.049] 0.040	0.96 [0.41] 0.014	0.46 [0.70] 0.007	6.30 [0.001] 0.092	4.24 [0.001] 0.062	2.60 [0.05] 0.039	5.84 [0.001] 0.085			Exp > Ctrl
RTC: Precontemplation	4.79 [0.030] 0.024	0.74 [0.389] 0.004	0.36 [0.550] 0.002	3.34 [0.069] 0.017	2.67 [0.10] 0.014	6.030 [0.015] 0.030	0.66 [0.42] 0.003	−1.95 (3.719)	−1.046 (3.270)	Exp < Ctrl
RTC: Contemplation	2.28 [0.133] 0.012	35 [0.556] 0.002	0.69 [0.408] 0.004	2.34 [0.13] 0.012	8.96 [0.003] 0.045	1.103 [0.29] 0.006	13.86 [0.000] 0.067	4.12 (3.33)	3.44 (3.60)	
RTC: Action	2.64 [0.106] 0.014	2.23 [0.137] 0.011	0.84 [0.361] 0.004	16.59 [0.001] 0.080	5.97 [0.015] 0.030	0.252 [0.62] 0.001	0.093 [0.76] 0.000	3.13 (3.45)	2.56 (3.39)	
** Model 6, F (8, 185) **	1.89 [0.064] 0.076	.63 [0.75] 0.027	1.087 [0.37] 0.045	1.64 [0.12] 0.066	0.73 [0.67] 0.031	2.17 [0.031] 0.086	1.32 [0.24] 0.054			
SCQ: Pleasant emotions	5.39 [0.021] 0.027	0.30 [0.58] 0.002	0.94 [0.33] 0.005	4.30 [0.039] 0.022	0.29 [0.59] 0.001	4.24 [0.041] 0.022	1.91 [0.16] 0.010	75.10 (20.93)	69.44 (21.52)	Exp > Ctrl
SCQ: Unpleasant emotions	0.76 [0.38] 0.004	0.66 [0.42] 0.003	0.025 [0.87] 0.001	0.59 [0.44] 0.003	0.657 [0.42] 0.003	0.073 [0.79] 0.001	0.76 [0.38] 0.004	66.95 (57.17)	60.56 (26.65)	
SCQ: Urge and temptation	7.42 [0.007] 0.037	0.29 [0.59] 0.001	1.20 [0.27] 0.006	7.35 [0.007] 0.037	0.019 [0.89] 0.000	4.971 [0.027] 0.025	0.170 [0.68] 0.001	62.88 (26.37)	54.49 (25.08)	Exp > Ctrl
SCQ: Positive social situations	1.92 [0.17] 0.010	0.37 [0.54] 0.002	0.55 [0.46] 0.003	9.83 [0.002] 0.049	2.15 [0.14] 0.011	0.16 [0.69] 0.001	0.10 [0.75] 0.001	64.67 (25.74)	57.13 (24.99)	
SCQ: Social tension	1.99 [0.160] 0.010	1.13 [0.29] 0.006	0.84 [0.36] 0.004	0.70 [0.40] 0.004	2.12 [0.15] 0.011	1.64 [0.20] 0.008	3.078 [0.081] 0.016	63.91 (30.92)	59.90 (28.00)	
SCQ: Social problems at work	4.04 [0.046] 0.021	0.008 [0.92] 0.001	0.047 [0.83] 0.001	0.62 [0.43] 0.003	1.56 [0.21] 0.008	0.003 [0.96] 0.000	2.38 [0.12] 0.012	73.038 (56.23)	58.33 (27.90)	Exp > Ctrl
SCQ: Testing self-control	0.91 [0.34] 0.005	0.076 [0.78] 0.001	0.004 [0.95] 0.001	9.67 [0.002] 0.048	1.23 [0.27] 0.006	2.061 [0.15] 0.11	3.050 [0.082] 0.016	64.13 (26.02)	60.65 (24.43)	
SCQ: Physical discomfort	3.39 [0.067] 0.017	0.10 [0.75] 0.001	1.16 [0.28] 0.006	1.68 [0.19] 0.009	0.82 [0.36] 0.004	8.06 [0.005] 0.040	4.00 [0.047] 0.020	65.87 (27.46)	61.39 (25.92)	

Note. RTC = Readiness to Change; SCQ = Situational Confidence Questionnaire.

## Data Availability

The datasets generated during and/or analyzed during the current study are available from the corresponding author upon reasonable request.

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
