# Peer review of "Do I Really Want to Change? The Effectiveness of Goal Ambivalence Feedback on Dieters’ Motivation"

_behavsci, 2022, doi:10.3390/bs12110441_

Round 1
Reviewer 1 Report
I recommend to the auhtors to split the research - presenting udner the form of 3 studies is a little bit unusual. On the other had, the auhtors could transform it into 3 hypothesis if they want to maintain the ideas. The manuscript should be re-organized to be suitable for publication as an article.
Author Response
Thank you for your comment. I understand that presenting study results in form of two or more studies is not common, but it is not also unusual. Like others, we have already published articles that present our research in sections identified as studies. Therefore, we would appreciate it if the reviewer could agree with the current format of our manuscript. Thank you.
Moreover, to improve the theoretical background and relevant evidence in the area we added the following section to the introduction: "The widely used Inter-goal Relations Questionnaire (IRQ; Riediger, 2001; Riediger & Freund, 2004) in health-related settings evaluates the extent to which pursuing a goal would contradict with or impose time, energy, and financial constraints for pursuing other goals or alternatively it would set up stage for and facilitate achieving other goals. Therefore, it seems that the IRQ evaluates inter-goal relationships mainly through their instrumentality dimension. Research exploring the impact of inter-goal facilitation / conflict and the pursuit of health-related goals has led to mixed results. Riediger et al. (2005) reported that inter-goal facilitation was related to the intensity of physical exercise, regardless of the age of the participants. Elliston et al (2020), administering the IRQ reported that inter-goal conflict and facilitation have not been able to predict dieters' weight loss in long-term via changes in their mood or snacking and food consumption."
Finally, the analyses section of our manuscript has been largely improved as a result of revisions requested by another reviewer. We attach the revised version of the manuscript with tracked changes.
Reviewer 2 Report
The authors present a two study paper – one in which they establish a novel scale to predict health behaviours by factoring in the commitment to competing goals. The area of goal competition is difficult to measure, and the novel scale does a good job of operationalising not only the health goal but competing goals in a short number of items.
Another between subjects experiment in which participants are given feedback in one group, based on this scale scores. The findings being the accounting for other goal commitments predicts dieting frequently more than just health intention, and the giving feedback on the potential dieting success increases intentions/beliefs/ attitudes towards dieting.
The paper is interesting, and contains some novel ideas in an appropriate samples. My main issues are related to study 2, as there are potential issues with the MANOVA analysis from lack of definition of the outcome variable, and the suitability of the control group. My review is dividing into methods, analysis, and interpretation sections below.
Methods/clarity
For the participants section it’s not clear what the demographics were of the final sample. Having demographic post exclusions would be clearer than having demographic details for the participants prior to exclusion.
For study 1, a bit more detail about the item evaluation would be better – i.e. how many participants were included in each sample? Were these participants similar in demographic details to the main sample?
In study 2, how exactly were the items altered in the changed version of the RTCQ and SCQ? Was it a simple change from alcohol to food wording, or something more elaborate?
In study 2 a table of group characteristics would be useful to interpret the groups, inclusion of p-values and effect sizes from the comparison would also assist interpretation.
Analysis
For the factor analysis, a bit more clarity regarding the method selected is needed to justify why PCA selected. If a planned factor structure was expected, a CFA would be a better approach – providing evidence of whether the theoretically derived factor fit the data. If it is more exploratory this would need to be outlined, then potentially using confirmatory analysis in study 2 data.
In study 2, the majority of assumptions for MANOVA are met – e.g. homogeneity of variance, however the DV’s for Model 1 aren’t correlated, with many being r < .1 and non-significant. This calls into question how conceptually similar the overall MANOVA outcome construct is. MANOVA is most ideally used when DVs are moderately correlated, reflecting different components of an underlying outcome factor. Discussion of the specific outcome factor is needed in more detail here to justify the use of MANOVA, as the study 2 outcome variable has multiple measures reflecting health intentions/behaviour which don’t have an immediately obvious underlying single factor. This is especially compared to the study 1 frequency of dieting outcome variable.
An interesting analysis which would confirm the pattern of results in study 1 with study 2 data would be correlate the GAS score with the other outcomes, rather than just using it as an experimental manipulation. It seems unusual to develop a novel scale and not explore it in the sample. Would the experimental effect potentially be dependent upon the GAS score?
An analysis which would improve the validity of the measure would be to assess its predictive validity over other competing scales. If the novel GAS is a better predictor (e.g. accounting for more variance) of behaviour or attitudes relative to other scales which don’t control for competing goal commitments, then this would be supportive of the scales validity and utility, relative to other established scales of health intentions. The authors do something similar by comparing the R-squared increase when adding Part B of the GAS to the model, but conducting this with established scales would improve the utility and impact of the novel scale. This may be a better approach for a future study though as it would require an additional sample.
Interpretation
A confound in the study is that the Expt group was given feedback, whilst the control group was not. It could be that any feedback regardless of content could have caused the effect – and that the specific content of the feedback based on goal-theory/ambivalence was not the determining factor. This is a considerable limiting factor which must at least be addressed in the limitations section.
One potential limitation which would prevent strong application of the current finding is that it is unclear how lasting the effect is – would this last beyond a follow-up? How much of this is demands characteristics due to wanting to appeal to the experimenters opinions?
Minor points
Typo’s – abstract: “mptivation”
Some font inconsistency
The terms part A/B and Part 1 or 2 are used interchangeably in table 2 – using just one is clearer.
Author Response
Reviewer 2
Comments and Suggestions for Authors
The authors present a two-study paper – one in which they establish a novel scale to predict health behaviours by factoring in the commitment to competing goals. The area of goal competition is difficult to measure, and the novel scale does a good job of operationalising not only the health goal but competing goals in a short number of items. Another between subjects experiment in which participants are given feedback in one group, based on this scale scores. The findings being the accounting for other goal commitments predicts dieting frequently more than just health intention, and the giving feedback on the potential dieting success increases intentions/beliefs/ attitudes towards dieting.
The paper is interesting and contains some novel ideas in an appropriate samples. My main issues are related to study 2, as there are potential issues with the MANOVA analysis from lack of definition of the outcome variable, and the suitability of the control group. My review is dividing into methods, analysis, and interpretation sections below.
Methods/clarity
- For the participants section it’s not clear what the demographics were of the final sample. Having demographic post exclusions would be clearer than having demographic details for the participants prior to exclusion.
*Authors' response
We do appreciate the reviewer's positive comment on the quality of our work and the valuable points that they have provided us to improve the manuscript. In response to the demographic information (Study 2), we added post exclusion demographic information to the section describing participants. Thank you.
- For study 1, a bit more detail about the item evaluation would be better – i.e., how many participants were included in each sample? Were these participants similar in demographic details to the main sample?
*Authors' response
We added more information regarding the samples involved in the item evaluation phase of the study; the section now reads, " Next, we conducted a pilot study with a few samples of dieters (n = 10 x2) and diabetics (n = 10) in clinical settings similar to the main study to ascertain the clarity and ease of understating the GAS items and instructions."
- In study 2, how exactly were the items altered in the changed version of the RTCQ and SCQ? Was it a simple change from alcohol to food wording, or something more elaborate?
*Authors' response
It was simple change from alcohol to food and drinks wording. We clarified the change in the manuscript by adding the following sentence to the text: "... Therefore, modification of the items was a simple wording change from alcohol to food and drinks."
- In study 2 a table of group characteristics would be useful to interpret the groups, inclusion of p-values and effect sizes from the comparison would also assist interpretation.
*Authors' response
We reported means, SDs, t-, and p-values in newly added Table 3.
Analysis
- For the factor analysis, a bit more clarity regarding the method selected is needed to justify why PCA selected. If a planned factor structure was expected, a CFA would be a better approach – providing evidence of whether the theoretically derived factor fit the data. If it is more exploratory this would need to be outlined, then potentially using confirmatory analysis in study 2 data.
*Authors' response
In addition to calculating an Exploratory Factor Analysis, using Principal Axis Factoring, we reported the results of a Confirmatory Factor Analysis (SEM, AMOS) for Study 1. We repeated the EFA and CFA with data from participants in the experimental group in Study 2. The new factor analyses supported the factor structure of the Goal Ambivalence Scale. The results are now added to the result sections for Study 1 and Study 2. The sections are note pasted in this letter because of the length of the sections.
- In study 2, the majority of assumptions for MANOVA are met – e.g., homogeneity of variance, however the DV’s for Model 1 aren’t correlated, with many being r < .1 and non-significant. This calls into question how conceptually similar the overall MANOVA outcome construct is. MANOVA is most ideally used when DVs are moderately correlated, reflecting different components of an underlying outcome factor. Discussion of the specific outcome factor is needed in more detail here to justify the use of MANOVA, as the study 2 outcome variable has multiple measures reflecting health intentions/behaviour which don’t have an immediately obvious underlying single factor. This is especially compared to the study 1 frequency of dieting outcome variable.
*Authors' response
We are grateful for this important comment. We conducted four different ANCOVAs instead of one MANCOVA for the reasons that the reviewer stated. The new results added to one more significant outcome for "Need for Information." Subsequently, we modified the text describing the models and Table 4 that presents the new analysis (Models 1-4). Thank you.
- An interesting analysis which would confirm the pattern of results in Study 1 with Study 2 data would be correlate the GAS score with the other outcomes, rather than just using it as an experimental manipulation. It seems unusual to develop a novel scale and not explore it in the sample. Would the experimental effect potentially be dependent upon the GAS score?
*Authors' response
As suggested by the reviewer, we calculated the correlations among the GAS Part A, Part B, and its Goal Commitment Likelihood Score (GCLS) with the study outcomes. Table 6 (new table) shows the correlation matrix, and the results were described in the text also.
- An analysis which would improve the validity of the measure would be to assess its predictive validity over other competing scales. If the novel GAS is a better predictor (e.g., accounting for more variance) of behaviour or attitudes relative to other scales which don’t control for competing goal commitments, then this would be supportive of the scale's validity and utility, relative to other established scales of health intentions. The authors do something similar by comparing the R-squared increase when adding Part B of the GAS to the model but conducting this with established scales would improve the utility and impact of the novel scale. This may be a better approach for a future study though as it would require an additional sample.
*Authors' response
- This is a very important point. Accordingly, in the final discussion, we added a new suggestion for future studies: "Additionally, administering competing measures of goal commitment would help further explore the predictive validity of the GAS relative to other established scales of health intentions."
Interpretation
- A confound in the study is that the Expt group was given feedback, whilst the control group was not. It could be that any feedback regardless of content could have caused the effect–and that the specific content of the feedback based on goal-theory/ambivalence was not the determining factor. This is a considerable limiting factor which must at least be addressed in the limitations section.
One potential limitation which would prevent strong application of the current finding is that it is unclear how lasting the effect is – would this last beyond a follow-up? How much of this is demands characteristics due to wanting to appeal to the experimenters' opinions?
*Authors' response
We added the following sentences to the text: "Moreover, the control group did not receive any form of feedback; therefore, observed changes could be attributable to the sheer feedback that they have received. To control for this potential issue, future studies may use sham feedback with the control sample." … " Finally, it is necessary that future studies explore long-term outcomes of providing patients with their GCLS in clinical and non-clinical settings."
Minor points
Typo’s – abstract: “mptivation” = corrected
Some font inconsistency = corrected
The terms part A/B and Part 1 or 2 are used interchangeably in table 2 – using just one is clearer. = corrected
Round 2
Reviewer 1 Report
The authors provided an improved version of the manuscript.
More details about results/statistical analysis were inserted.
However, due to its use in several published researches - as the authors replay - please take into consideration slice-salamy before publishing.